# Differential Enrichment of Trace and Major Elements in Biodegraded Oil: A Case Study from Bohai Bay Basin, China

Haifeng Yang [1], Deying Wang [1], Feilong Wang [1], Yanfei Gao [1], Guomin Tang [1], Youjun Tang [2,3] and Peng Sun [2,3,*]

[1]    Tianjin Branch of China National Offshore Oil Corporation, Tianjin 300452, China
[2]    Key Laboratory of Exploration Technologies for Oil and Gas Resources, Ministry of Education, Yangtze University, Wuhan 430100, China
[3]    School of Petroleum Engineering, Yangtze University, Wuhan 430100, China
*    Correspondence: p_sun@yangtzeu.edu.cn

**Abstract:** Inorganic elements in crude oil have been used in the reconstruction of the sedimentary environment and oil–oil (source) correlations; however, the effect of biodegradation on these elements has not been investigated sufficiently. In this study, 14 crude oils from the Miaoxi Sag of the Bohai Bay Basin, eastern China, were analyzed using molecular markers, trace elements, and major elements to determine the effect of biodegradation on inorganic elements. The molecular markers indicated that the oils are in the low maturity stage and are derived from similar parent materials in lacustrine source rocks. The high-sulfur oil came from a more reductive and saltier environment compared with the low-sulfur oil. The oils were subjected to varying degrees of biodegradation. The concentrations of Mg, Ca, Mn, Fe, Be, Sc, Rb, Sr, Zr, Pb, Th, and U increased significantly throughout the biodegradation process, while the concentrations of Na, K, Ti, Al, Cr, Zn, Cs, Nb, Ba, Hf, and Tl increased considerably only during the intense biodegradation stage (PM < 4). The concentrations of P, Li, V, Co, Ni, Cu, Ga, Sn, and Ta were not correlated with the level of biodegradation. The V/Ni, V/Co, Ni/Co, Cr/V, Sc/V, and Th/U ratios were affected by biodegradation when PM $\geq$ 4. Several ratios, including Mg/P, Ca/P, Mn/P, and Fe/P, are proposed as favorable indicators of the level of biodegradation. Differential enrichment of these elements is associated with the effects of organic acids generated by biodegradation on the oil–water–rock interactions in the reservoir.

**Keywords:** major elements; trace elements; biodegradation; organic acid; oil–water–rock interaction

## 1. Introduction

Inorganic elements have been widely used in the correlation of oil–oil (source) and the reconstruction of sedimentary environments, making them valuable tools for research on petroleum systems [1–3]. Among them, the trace elements V and Ni are most commonly used. V and Ni are present primarily in crude oil as metalloporphyrin and non-porphyrin complexes [4–6]. This unique configuration enables V and Ni to preserve information on source rocks during petroleum formation, migration, and accumulation [7–10]. Therefore, the V/Ni ratio has been widely used in genetic correlation and as an indicator of the sedimentary environment [3,4,10,11]. Other parameters such as V/Co, Ni/Co, Cr/V, Sc/V, and Th/U ratios have also been discovered and used in several complex petroleum systems in the Sichuan basin, the Tarim basin, and the Bohai Bay basin of China [3,11–14].

The geochemical behaviors of different elements in crude oil are not well understood, and the effect of biodegradation on inorganic elements is not clear. Biodegradation is the most common type of secondary alteration of crude oil [15–17], and approximately 2/5 of the petroleum in the world has been damaged or altered by bacteria [18]. Because biodegradation can change the chemical composition and physical properties of crude oil, evaluating the degree of biodegradation is important for petroleum exploration and development [19,20]. Currently, the degree of biodegradation is mainly determined using

organic molecular markers [21–25], which are effective for crude oils with a low degree of degradation. However, the organic molecular markers of heavily or ultra-heavily biodegraded oils are severely consumed or completely degraded, making accurate assessment of the biodegradation level using this method difficult. For example, most crude oil samples from the Peace River oil sands in Alberta have been degraded to PM5, but their viscosity varies significantly [26]. The abundant inorganic elements in such crude oil may offer a potential solution, but current research is generally limited to V and Ni. Bioenrichment can increase the quantity of V and Ni complexes, improving their resistance to bacteria [27,28]. As a result, when organic molecular markers are severely degraded, the concentrations of V and Ni increase and can therefore be used as geochemical markers of highly degraded oils [2,29]. However, their utility as biodegradation indicators seems to be conditional in our study.

In this study, 14 crude oil samples from the Miaoxi Sag in the Bohai Bay Basin, eastern China, were collected and analyzed using organic molecular markers and major and trace inorganic elements. The maturity, origin, and biodegradation levels of the oil samples were evaluated using organic molecular markers. Differential enrichment of elements during the biodegradation process was studied. This study helps to improve our understanding of the behavior of inorganic elements during biodegradation.

## 2. Geological Setting

The Bohai Bay basin is a fault-depression basin in eastern China that developed in the basement of the Paleozoic North China crater (Figure 1a). The Miaoxi Sag is located in the eastern sea area of the Bohai Bay Basin (Figure 1b) and is a secondary structural unit of the Miaoxi Depression, with an area of 1090 km². The Sag extends in the direction of NNE and is adjacent to the Bonan and Laibei subuplifts in the west, the Jiaoliao uplift in the east, and the Miaoxinan subuplift in the north (Figure 1c).

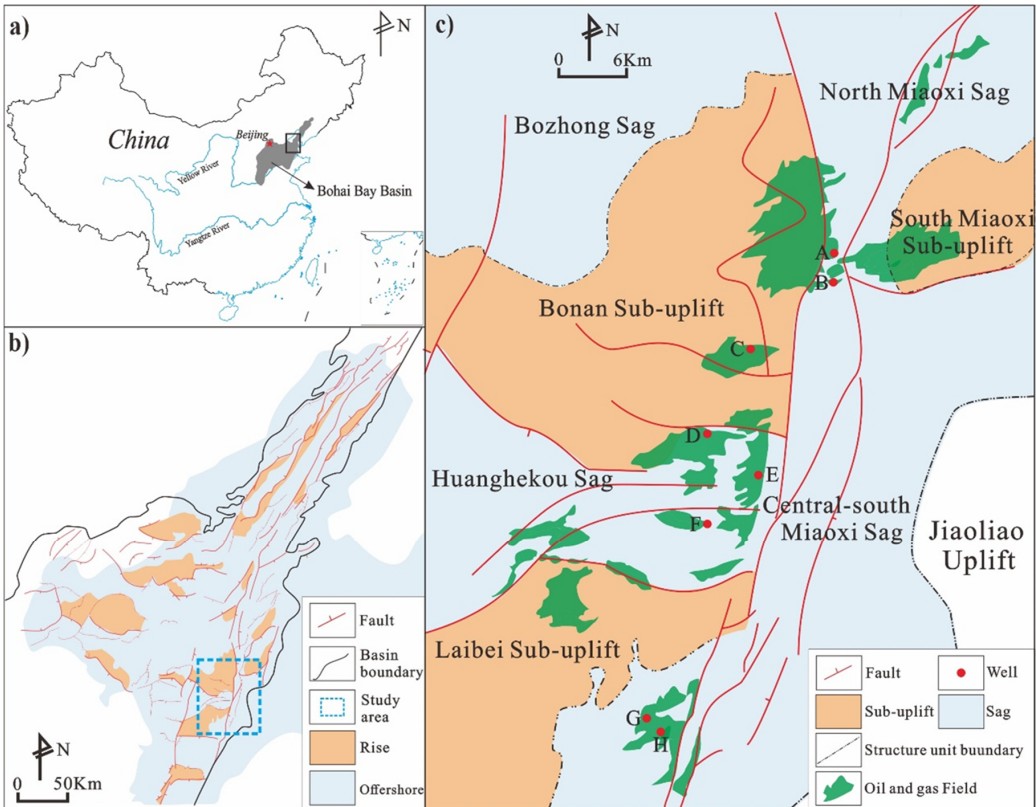

**Figure 1.** Maps showing the locations of the Bohai Bay Basin. (**a**) The study area (**b**) and the distribution of sampled well (**c**).

From the bottom to the top, the Cenozoic strata drilled for exploration wells include the Kongdian Formation (E*k*), the Shahejie Formation (E*s*), the Dongying Formation (E*d*), the Guantao Formation (N$_1$*g*), and the Minghuazhen Formation (N$_1$*m*). E*s* is further divided into four sections: E*s*$^4$, E*s*$^3$, E*s*$^2$, and E*s*$^1$. Major petroleum reservoirs have been discovered in N$_1$*g* and N$_1$*m*, while minor reservoirs have been discovered in E*s*$^2$ and E*d*. Previous studies have revealed that four sets of source rocks, i.e., dark mudstones E*s*$^4$, E*s*$^3$, E*s*$^{1-2}$, and E*d*$^3$, developed in the eastern sea area of the Bohai Bay Basin. The oil in Miaoxi Sag originated mainly from E*s*$^3$ and E*s*$^4$ dark mudstones [30–32].

## 3. Samples and Methods

### 3.1. Samples

A total of 14 crude oil samples from the central and southern parts of the Miaoxi Sag, Bohai Bay Basin (Table 1, Figure 1c) were analyzed. The crude oil samples were collected from 8 wells in N$_1$*m*, N$_1$*g*, E*d*, and E*s* at depths ranging from 1040 to 2600 m; some oils were collected from different depths of the same well. Based on their sulfur content, these crude oils can be classified into low-sulfur oils (0.31% to 0.52%) and high-sulfur oils (1.07% to 2.80%) (Table 1).

**Table 1.** Basic information and geochemical parameters based on the steranes and terpanes in the studied oil samples.

| Sample ID | Well | Depth (m) | Strata | S (%) | T1 | T2 | T3 | T4 | S1 | S2 | S3 | S4 | S5 | S6 | S7 | S8 | Oil Types |
|---|---|---|---|---|---|---|---|---|---|---|---|---|---|---|---|---|---|
| A-1 | A | 1191–1207 | N$_1$ | 0.40 | 0.57 | 0.58 | 0.36 | 0.44 | 1.29 | 0.23 | 32.27 | 30.02 | 37.71 | 34.59 | 29.40 | 36.01 | |
| A-2 | A | 1267–1317 | N$_1$ | 0.44 | 0.57 | 0.55 | 0.43 | 0.39 | 1.25 | 0.23 | 32.79 | 29.65 | 37.56 | 36.04 | 28.20 | 35.76 | |
| A-3 | A | 1449–1472 | N$_1$ | 0.50 | 0.57 | 0.56 | 0.41 | 0.38 | 1.31 | 0.23 | 31.99 | 29.55 | 38.46 | 37.56 | 27.53 | 34.92 | |
| B-1 | B | 1284 | N$_1$*g* | 0.45 | 0.57 | 0.57 | 0.38 | 0.32 | 1.39 | 0.19 | 26.75 | 32.86 | 40.39 | 33.30 | 26.25 | 40.45 | |
| B-2 | B | 1417–1441 | N$_1$*g* | 0.50 | 0.56 | 0.55 | 0.35 | 0.29 | 1.33 | 0.22 | 30.69 | 29.54 | 39.77 | 36.47 | 27.60 | 35.93 | Low-sulfur |
| C-1 | C | 1040 | N$_1$*g* | 0.44 | 0.61 | 0.57 | 0.50 | 0.42 | 1.31 | 0.15 | 29.98 | 31.63 | 38.39 | 28.20 | 32.46 | 39.34 | |
| G-1 | G | 2561–2565 | E*s* | 0.43 | 0.57 | 0.56 | 0.35 | 0.30 | 1.39 | 0.21 | 29.40 | 29.14 | 41.46 | 37.06 | 25.19 | 37.75 | |
| G-2 | G | 2585–2603 | E*s* | 0.52 | 0.57 | 0.56 | 0.36 | 0.32 | 1.39 | 0.21 | 28.96 | 28.43 | 42.61 | 37.15 | 25.99 | 36.86 | |
| H-1 | H | 2153–2181 | E*d* | 0.31 | 0.57 | 0.56 | 0.37 | 0.32 | 1.30 | 0.22 | 33.28 | 29.82 | 36.90 | 36.98 | 25.51 | 37.50 | |
| D-1 | D | 1297–1322 | N$_1$*m* | 2.80 | 0.57 | 0.59 | 0.42 | 0.38 | 1.10 | 0.25 | 31.59 | 28.13 | 40.28 | 32.58 | 27.79 | 39.63 | |
| D-2 | D | 1552–1570 | N$_1$*g* | 2.07 | 0.57 | 0.57 | 0.38 | 0.35 | 1.19 | 0.24 | 32.32 | 28.09 | 39.59 | 34.82 | 27.23 | 37.95 | |
| E-1 | E | 1507–1529.5 | N$_1$*g* | 1.12 | 0.57 | 0.58 | 0.42 | 0.37 | 1.29 | 0.24 | 33.31 | 29.37 | 37.32 | 34.38 | 27.28 | 38.34 | High-sulfur |
| E-2 | E | 1241–1259 | N$_1$*m* | 1.07 | 0.58 | 0.61 | 0.45 | 0.40 | 1.25 | 0.26 | 35.04 | 29.23 | 35.73 | 33.19 | 27.48 | 39.34 | |
| F-1 | F | 2132 | E*d* | 1.32 | 0.58 | 0.59 | 0.47 | 0.39 | 1.24 | 0.22 | 25.64 | 32.80 | 41.56 | 31.49 | 27.44 | 41.07 | |

Note: T1 = 22S/(22S + 22R)-C$_{31}$H; T2 = 22S/(22S + 22R)-C$_{32}$H; T3 = 20S/(20S + 20R)-C$_{29}$RS; T4 = $\beta\beta$/($\beta\beta$ + $\alpha\alpha$)-C$_{29}$RS; S1 = C$_{26}$/C$_{25}$TT; S2 = C$_{31}$R/C$_{30}$H; S3, S4, and S5 = relative contents (%) of C$_{19}$ + C$_{20}$TT, C$_{21}$ TT, and C$_{23}$TT, respectively; S6, S7, and S8 = relative contents (%) of $\alpha\alpha\alpha$(20R)-C$_{27}$RS, $\alpha\alpha\alpha$(20R)-C$_{28}$RS, and $\alpha\alpha\alpha$(20R)-C$_{29}$RS, respectively; H = hopane; RS = regular steranes; TT = tricyclic terpane.

### 3.2. Oil Analysis

Classical silica gel chromatography was used to separate saturated hydrocarbon components from crude oil for gas chromatography–mass spectrometry (GC–MS) analysis. Excess *n*-pentane was added to approximately 50 mg of oil samples and dissolved, precipitated, and filtered to remove asphaltene from the oil. The rest was injected into the chromatography column filled with alumina and silica gel. Saturated hydrocarbons, aromatic hydrocarbons, and non-hydrocarbons were then separated by adding *n*-hexane (50 mL), toluene (50 mL), and a chloroform/methanol combination (70 mL, 98:2), respectively. An Agilent 6890A GC–Agilent 5975I MS was used to identify saturated hydrocarbon components. The GC was equipped with an HP–5 MS column (60 m × 0.25 mm × 0.25 μm). The GC temperature was set to 50 °C for 1 min, increased to 250 °C at 3 °C/min and then to 310 °C at 20 °C/min, where it was maintained for 10 min.

The crude oil samples were pretreated using an acid digestion method to determine the inorganic elements. The following specific operations were conducted: approximately 25 mg of the crude oil sample was transferred to a sealable Teflon cup, treated with HNO$_3$ (2 mL), HF (3 mL), and HCl (1 mL) and then dried on a hot plate (150 °C). Subsequently, the sample was treated with HNO$_3$ (1 mL) and HF (3 mL) and placed in an oven (180 °C)

for 48 h to dissolve. After cooling to room temperature, $HClO_4$ (0.5 mL) was added to the solution and then dried on a hot plate (150 °C). After cooling, $HNO_3$ (2 mL) was added to the solution and the mixture placed in a drying oven (150 °C) for 12 h. Finally, the solution was transferred to a volumetric flask (50 mL) containing ultrapure water. The elements were measured using inductively coupled plasma mass spectrometry (ICP-MS). Rh was employed as an internal standard to quantify the concentrations of the elements.

## 4. Results

### 4.1. Molecular Markers

There were significant differences in the distribution patterns of *n*-alkanes in the crude oil samples: from none to slight to partial or complete consumption of *n*-alkanes (Figure 2). Among terpenoids, tricyclic terpanes (TT) were present in relatively low abundance, although the whole sequence from $C_{19}TT$ to $C_{30}TT$ were detected, with $C_{23}TT$ being dominant (Figure 3). Compared to the TTs, hopanes (H) were present in higher abundances, with $C_{30}H$ being dominant. Elements with high carbon numbers—$C_{31}H$ to $C_{35}H$—were identified (Figure 3). Furthermore, 25-norhopanes (NH) were identified in several oil samples, but their distributions varied significantly. For example, no 25-norhopane was detected in sample E-1, while a complete series of 25-norhopanes was detected in sample A-1 (Figure 3). Among the steranes, $C_{27}$, $C_{28}$, and $C_{29}$ regular steranes (RS) were the most abundant and displayed a "V" type, while pregnenes (PS) and disteranes (DS) of $C_{21}$–$C_{22}$ were present in relatively low abundance (Figure 3).

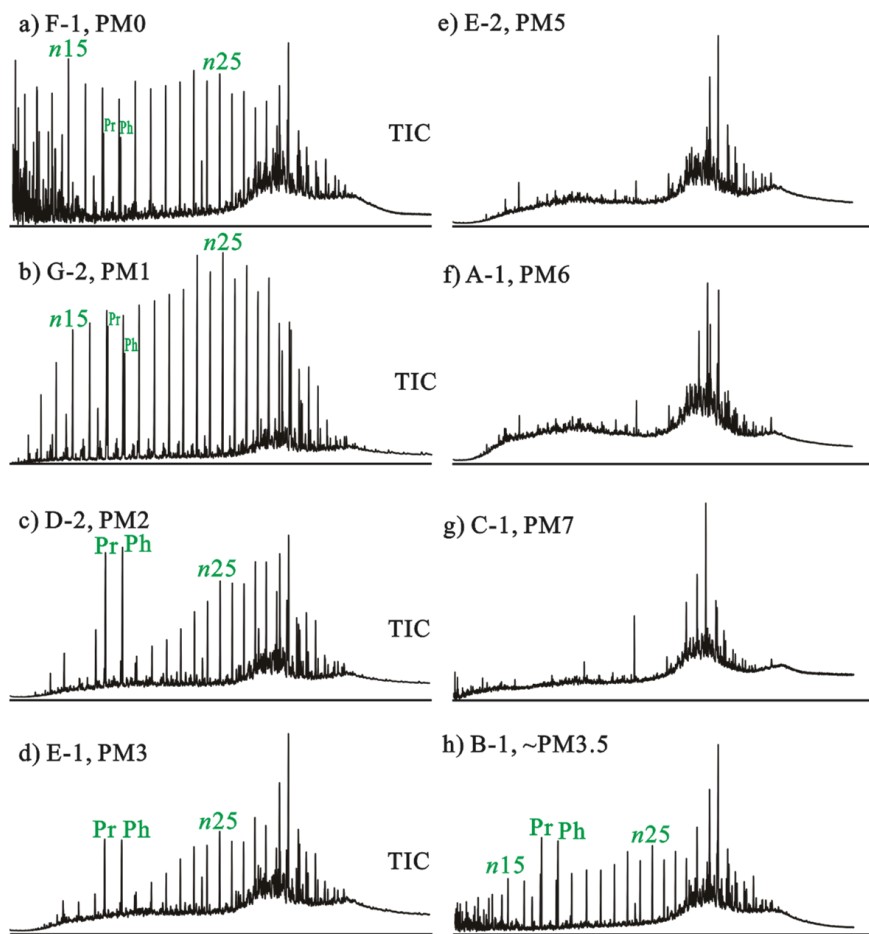

**Figure 2.** TIC chromatograms of selected oils with different degrees of biodegradation (PM Scale). Pr: Pristane; Ph: Phytane.

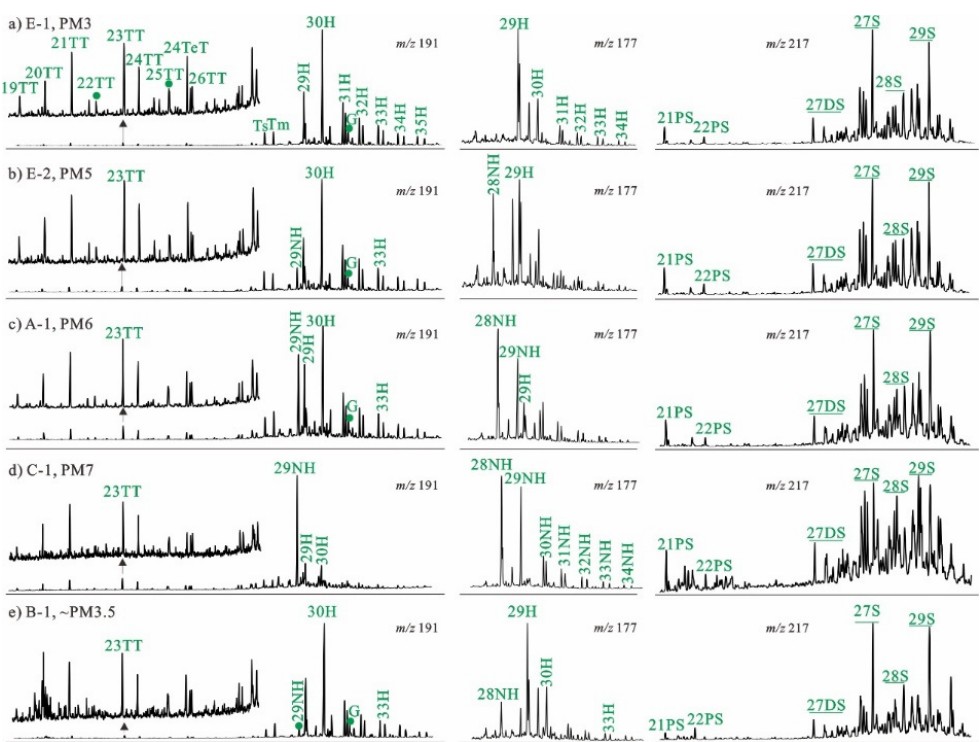

**Figure 3.** Mass chromatograms of selected oils with different degrees of biodegradation (PM Scale) showing the distribution of terpanes ($m/z$ 191), 25-norhopanes ($m/z$ 177), and steranes ($m/z$ 217). TT: tricyclic terpane; TeT: tetracyclic terpane; Ts: 18α(H)-trisnorneohopane; Tm: 17α(H)-trisnorhopane; H: Hopane; G: gammacerane; NH: 25-norhopane; 21-22S: PS: pregnene; DiaS: diasterane; RS: regular sterane; Numbers: carbon numbers.

Sample C-1 differed significantly from the other samples. In addition to its high consumption of hopanes, regular steranes also exhibited some alterations, while pregnenes were relatively abundant (Figure 3). Some important parameters calculated from the molecular markers are shown in Table 1.

*4.2. Inorganic Elements*

The major elements, including Na, K, Mg, Ca, Ti, Mn, Fe, Al, and P, were detected in crude oil samples at a total concentration of 152.43–10,787.68 μg/g (Figure 4, Table 2). The concentrations of the elements differed significantly: The concentrations of Na, K, Ca, and Mg were 48.04–4804.07 μg/g, 42.66–4064.12 μg/g, 11.05–1175.60 μg/g, and 4.48–322.96 μg/g, respectively. The concentrations of Fe, Al, Ti, and Mn were 14.13–257.21 μg/g, 5.53–122.47 μg/g, 0.86–7.21 μg/g, and 0.24–19.18 μg/g, respectively (Table 2). The concentration of non-metallic P was 9.25–20.18 μg/g. The concentration of these elements also varied in different samples. For example, the elements were more enriched in sample C-1 compared with the other samples. However, the concentrations of inorganic elements in different oil samples had similar distribution curves (Figure 4b).

The trace elements detected in the crude oil samples are shown in Figure 5. The total concentration of the trace elements (ΣTE) varied from 19.63 to 630.54 μg/g (Table 2), which was significantly less than the total concentration of the major elements (ΣME). The ΣTE/ΣME value was 0.022–0.327. The distribution of the trace elements in different samples were similar (Figure 5b). The ratios of the inorganic elements V/Ni, V/Co, Ni/Co, Cr/V, Sc/V, and Th/U were 0.026–0.112, 0.188–7.665, 6.958–92.165, 0.117–1.032, 0.0004–0.298, and 0.844–75.180, respectively (Table 3).

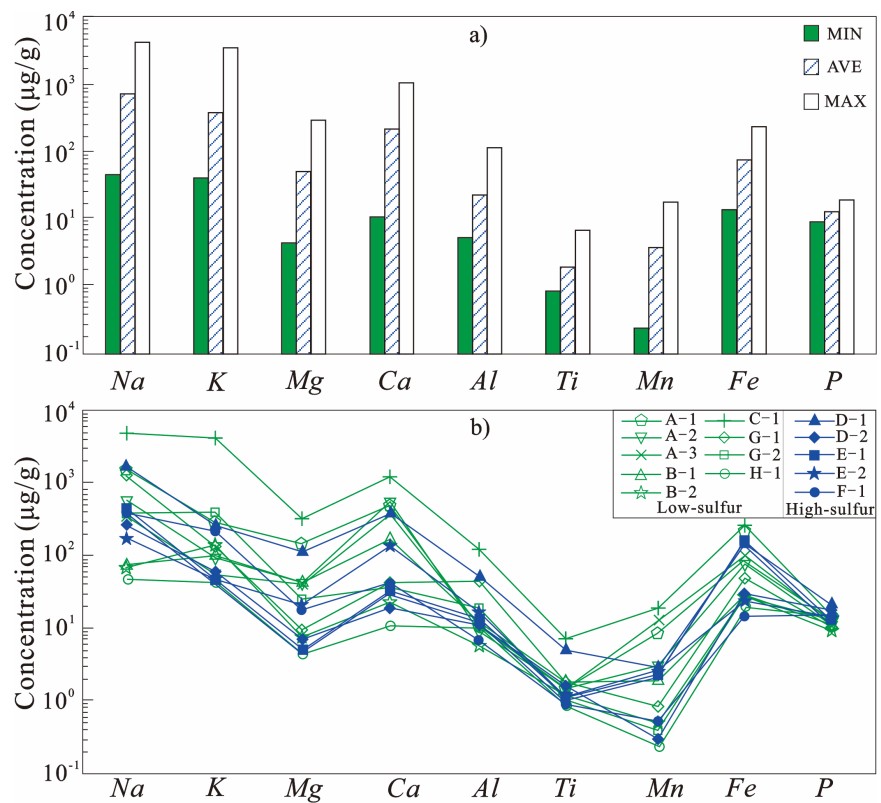

**Figure 4.** Concentrations (**a**) and distribution patterns (**b**) of major elements in the studied oil samples.

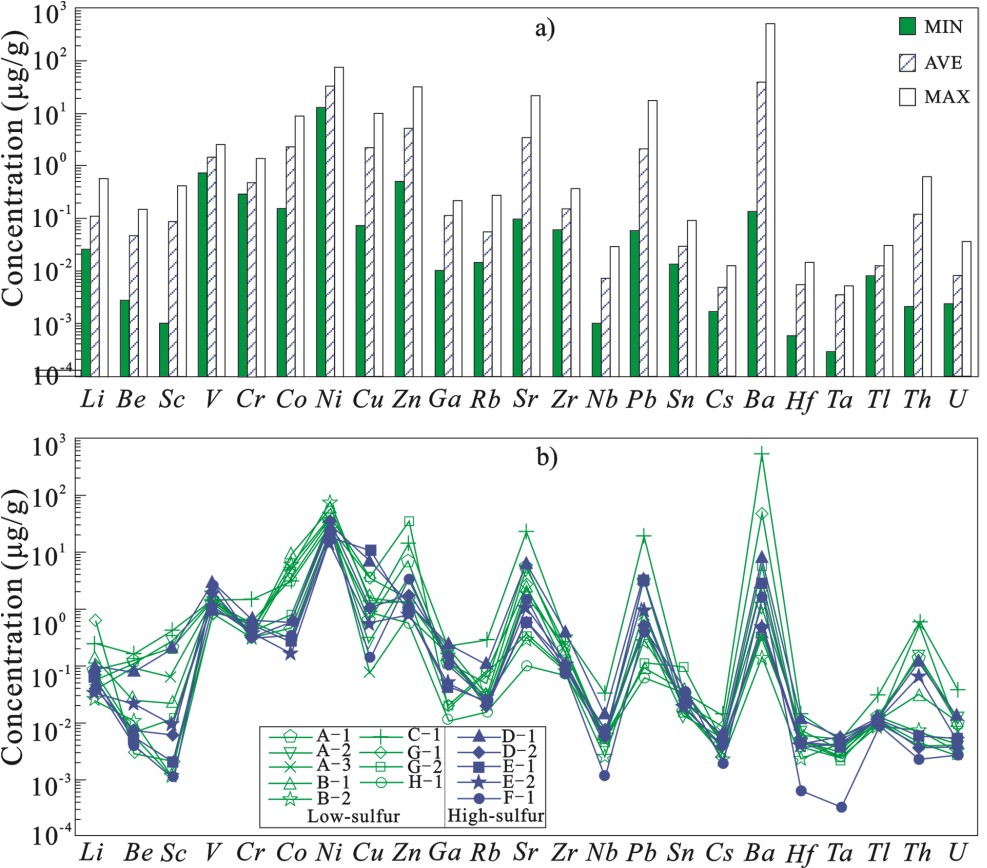

**Figure 5.** Concentrations (**a**) and distribution patterns (**b**) of trace elements in the studied oil samples.

**Table 2.** Concentrations of major and trace elements in the studied oil samples.

| Elements (µg/g) | | Sample ID | | | | | | | | | | | | | |
|---|---|---|---|---|---|---|---|---|---|---|---|---|---|---|---|
| | | A-1 | A-2 | A-3 | B-1 | B-2 | C-1 | G-1 | G-2 | H-1 | D-1 | D-2 | E-1 | E-2 | F-1 |
| Major elements | Na | 1462.45 | 561.26 | 338.91 | 71.35 | 68.71 | 4804.07 | 1259.87 | 376.31 | 48.04 | 1564.79 | 262.90 | 441.51 | 169.38 | 375.63 |
| | K | 280.14 | 91.44 | 56.63 | 97.43 | 137.30 | 4064.12 | 130.63 | 388.10 | 42.66 | 251.75 | 58.16 | 45.43 | 46.71 | 213.45 |
| | Mg | 143.33 | 40.96 | 40.94 | 41.50 | 7.39 | 322.96 | 9.50 | 24.68 | 4.48 | 108.45 | 6.93 | 5.02 | 20.34 | 17.71 |
| | Ca | 471.81 | 532.14 | 366.70 | 164.71 | 22.54 | 1175.60 | 42.25 | 36.09 | 11.05 | 352.65 | 18.77 | 32.07 | 128.50 | 41.27 |
| | Al | 10.58 | 8.86 | 11.37 | 12.15 | 5.53 | 122.47 | 43.77 | 18.49 | 10.17 | 49.68 | 10.94 | 13.10 | 16.68 | 6.83 |
| | Ti | 1.49 | 1.40 | 1.56 | 1.84 | 1.11 | 7.21 | 1.81 | 1.03 | 0.86 | 4.87 | 1.61 | 1.11 | 1.23 | 0.88 |
| | Mn | 8.28 | 2.94 | 12.89 | 1.89 | 0.49 | 19.18 | 0.83 | 0.40 | 0.24 | 2.79 | 0.30 | 2.31 | 2.50 | 0.52 |
| | Fe | 211.82 | 73.24 | 100.31 | 26.80 | 26.38 | 257.21 | 47.16 | 25.80 | 20.02 | 142.01 | 29.15 | 159.78 | 22.86 | 14.13 |
| | P | 12.56 | 10.62 | 11.98 | 10.55 | 9.25 | 14.86 | 10.46 | 13.07 | 14.91 | 20.18 | 17.24 | 13.48 | 13.92 | 14.98 |
| | ΣME | 2602.46 | 1322.85 | 941.29 | 428.21 | 278.69 | 10,787.68 | 1546.28 | 883.97 | 152.43 | 2497.17 | 406.00 | 713.80 | 422.13 | 685.40 |
| Trace elements | Li | 0.083 | 0.055 | 0.038 | 0.142 | 0.026 | 0.239 | 0.601 | 0.071 | 0.056 | 0.088 | 0.040 | 0.060 | 0.032 | 0.044 |
| | Be | 0.124 | 0.107 | 0.093 | 0.024 | 0.011 | 0.151 | 0.003 | 0.006 | 0.005 | 0.073 | 0.007 | 0.005 | 0.020 | 0.004 |
| | Sc | 0.206 | 0.275 | 0.062 | 0.022 | 0.001 | 0.423 | 0.002 | 0.011 | 0.001 | 0.190 | 0.006 | 0.002 | 0.009 | 0.001 |
| | V | 1.339 | 1.425 | 1.290 | 1.690 | 2.065 | 1.419 | 0.782 | 1.350 | 0.839 | 2.624 | 2.006 | 0.950 | 1.134 | 2.498 |
| | Cr | 0.571 | 0.398 | 0.357 | 0.384 | 0.312 | 1.464 | 0.298 | 0.383 | 0.551 | 0.616 | 0.297 | 0.512 | 0.364 | 0.293 |
| | Co | 3.228 | 4.582 | 5.675 | 8.977 | 5.196 | 2.967 | 0.443 | 0.754 | 0.265 | 0.529 | 0.529 | 0.239 | 0.156 | 0.326 |
| | Ni | 33.511 | 37.229 | 41.145 | 62.463 | 79.715 | 26.490 | 25.070 | 39.602 | 24.378 | 25.603 | 34.190 | 16.292 | 14.021 | 22.207 |
| | Cu | 0.636 | 0.257 | 0.073 | 1.594 | 1.361 | 0.825 | 3.348 | 3.497 | 0.832 | 6.822 | 0.947 | 10.174 | 0.512 | 0.134 |
| | Zn | 6.962 | 3.544 | 0.758 | 1.274 | 1.354 | 14.157 | 1.494 | 33.851 | 0.535 | 1.184 | 1.698 | 0.838 | 0.751 | 3.196 |
| | Ga | 0.115 | 0.120 | 0.162 | 0.212 | 0.150 | 0.216 | 0.019 | 0.015 | 0.010 | 0.229 | 0.159 | 0.039 | 0.049 | 0.097 |
| | Rb | 0.049 | 0.028 | 0.020 | 0.028 | 0.025 | 0.284 | 0.079 | 0.073 | 0.014 | 0.093 | 0.018 | 0.023 | 0.020 | 0.022 |
| | Sr | 5.527 | 5.061 | 1.639 | 2.343 | 0.320 | 22.928 | 2.613 | 0.345 | 0.094 | 5.964 | 0.578 | 0.610 | 1.005 | 1.402 |
| | Zr | 0.175 | 0.229 | 0.223 | 0.153 | 0.080 | 0.338 | 0.081 | 0.084 | 0.062 | 0.379 | 0.102 | 0.077 | 0.096 | 0.069 |
| | Nb | 0.005 | 0.005 | 0.005 | 0.005 | 0.002 | 0.030 | 0.007 | 0.004 | 0.005 | 0.014 | 0.005 | 0.006 | 0.005 | 0.001 |
| | Pb | 3.168 | 0.445 | 0.819 | 0.091 | 0.282 | 19.065 | 0.420 | 0.105 | 0.058 | 3.171 | 0.483 | 2.718 | 0.915 | 0.371 |
| | Sn | 0.016 | 0.013 | 0.017 | 0.037 | 0.026 | 0.034 | 0.021 | 0.091 | 0.030 | 0.025 | 0.021 | 0.023 | 0.017 | 0.038 |
| | Cs | 0.004 | 0.005 | 0.003 | 0.005 | 0.002 | 0.012 | 0.008 | 0.004 | 0.004 | 0.005 | 0.004 | 0.005 | 0.003 | 0.002 |
| | Ba | 2.452 | 1.126 | 0.313 | 0.363 | 0.135 | 538.765 | 45.954 | 5.493 | 0.349 | 6.896 | 0.448 | 2.649 | 0.437 | 1.536 |
| | Hf | 0.005 | 0.007 | 0.006 | 0.005 | 0.002 | 0.014 | 0.005 | 0.004 | 0.004 | 0.011 | 0.004 | 0.004 | 0.003 | 0.001 |
| | Ta | 0.003 | 0.003 | 0.005 | 0.003 | 0.004 | 0.005 | 0.004 | 0.002 | 0.002 | 0.005 | 0.005 | 0.004 | 0.004 | 0.000 |
| | Tl | 0.014 | 0.012 | 0.011 | 0.011 | 0.009 | 0.032 | 0.012 | 0.010 | 0.009 | 0.012 | 0.011 | 0.011 | 0.011 | 0.008 |
| | Th | 0.136 | 0.161 | 0.522 | 0.031 | 0.007 | 0.650 | 0.006 | 0.004 | 0.004 | 0.109 | 0.003 | 0.006 | 0.062 | 0.002 |
| | U | 0.009 | 0.011 | 0.007 | 0.010 | 0.003 | 0.037 | 0.004 | 0.002 | 0.004 | 0.011 | 0.004 | 0.005 | 0.004 | 0.003 |
| | ΣTE | 58.131 | 54.825 | 53.179 | 79.845 | 91.087 | 630.121 | 81.271 | 85.749 | 28.111 | 54.464 | 41.561 | 35.248 | 19.622 | 32.253 |

Note: ΣME: total concentration of major elements (µg/g); ΣTE: total concentration of trace elements (µg/g).

**Table 3.** Ratios of major and trace elements in the studied oil samples.

| Sample ID | ΣTE/ΣME | V/Ni | V/Co | Ni/Co | Cr/V | Sc/V | Th/U | Mg/P | Ca/P | Mn/P | Be/V | V/P | Ni/P | Co/P | Ga/P |
|---|---|---|---|---|---|---|---|---|---|---|---|---|---|---|---|
| A-1 | 0.022 | 0.040 | 0.415 | 10.381 | 0.426 | 0.154 | 14.396 | 11.410 | 37.560 | 0.659 | 0.093 | 0.107 | 2.668 | 0.257 | 0.009 |
| A-2 | 0.042 | 0.038 | 0.311 | 8.126 | 0.279 | 0.193 | 14.024 | 3.855 | 50.083 | 0.276 | 0.075 | 0.134 | 3.504 | 0.431 | 0.011 |
| A-3 | 0.057 | 0.031 | 0.227 | 7.250 | 0.277 | 0.048 | 75.180 | 3.417 | 30.608 | 1.076 | 0.072 | 0.108 | 3.434 | 0.474 | 0.014 |
| B-1 | 0.187 | 0.027 | 0.188 | 6.958 | 0.227 | 0.013 | 3.171 | 3.934 | 15.613 | 0.179 | 0.014 | 0.160 | 5.921 | 0.851 | 0.020 |
| B-2 | 0.327 | 0.026 | 0.397 | 15.341 | 0.151 | 0.000 | 2.402 | 0.799 | 2.437 | 0.053 | 0.005 | 0.223 | 8.619 | 0.562 | 0.016 |
| C-1 | 0.058 | 0.054 | 0.478 | 8.929 | 1.032 | 0.298 | 17.646 | 21.740 | 79.134 | 1.291 | 0.107 | 0.095 | 1.783 | 0.200 | 0.015 |
| G-1 | 0.053 | 0.031 | 1.766 | 56.622 | 0.381 | 0.002 | 1.658 | 0.908 | 4.038 | 0.079 | 0.004 | 0.075 | 2.396 | 0.042 | 0.002 |
| G-2 | 0.097 | 0.034 | 1.792 | 52.555 | 0.284 | 0.008 | 1.804 | 1.888 | 2.761 | 0.030 | 0.005 | 0.103 | 3.029 | 0.058 | 0.001 |
| H-1 | 0.184 | 0.034 | 3.172 | 92.165 | 0.657 | 0.001 | 0.972 | 0.300 | 0.741 | 0.016 | 0.006 | 0.056 | 1.635 | 0.018 | 0.001 |
| D-1 | 0.022 | 0.102 | 4.957 | 48.361 | 0.235 | 0.072 | 9.702 | 5.373 | 17.474 | 0.138 | 0.028 | 0.130 | 1.269 | 0.026 | 0.011 |
| D-2 | 0.102 | 0.059 | 3.793 | 64.638 | 0.148 | 0.003 | 0.957 | 0.402 | 1.089 | 0.017 | 0.003 | 0.116 | 1.983 | 0.031 | 0.009 |
| E-1 | 0.049 | 0.058 | 3.979 | 68.253 | 0.539 | 0.002 | 1.109 | 0.372 | 2.380 | 0.171 | 0.005 | 0.070 | 1.209 | 0.018 | 0.003 |
| E-2 | 0.047 | 0.081 | 7.253 | 89.702 | 0.321 | 0.008 | 14.305 | 1.462 | 9.234 | 0.180 | 0.018 | 0.081 | 1.008 | 0.011 | 0.004 |
| F-1 | 0.047 | 0.112 | 7.665 | 68.157 | 0.117 | 0.000 | 0.844 | 1.182 | 2.756 | 0.035 | 0.001 | 0.167 | 1.483 | 0.022 | 0.006 |

Note: ΣME: total concentration of major elements ($\mu g/g$); ΣTE: total concentration of trace elements ($\mu g/g$).

## 5. Discussion

### 5.1. Thermal Maturity

Crude oil is a mixture of various compounds. Its maturity can be characterized by the response of certain compounds to thermal evolution [33]. For example, the contents of hopanes and regular steranes decrease with increasing thermal evolution of source rocks, while the contents of TTs and diasteranes increase [34,35]. In these oil samples, the contents of hopanes and regular steranes are much higher than those of TTs and diasteranes (Figure 3), indicating relatively lower thermal maturity.

Several maturity parameters based on terpanes and hopanes have been widely used to assess crude oil maturity [35,36]. During the early stages of oil production, the values of $22S/(22S + 22R)$ of $C_{31}H$ and $C_{32}H$ increase with increasing maturity and reach the equilibrium point of thermal evolution when the values range from 0.57 to 0.62 [37]. Similarly, the values of $20S/(20S + 20R)$ and $\beta\beta/(\alpha\alpha + \beta\beta)$ of $C_{29}$ steranes also increase with increasing maturity, and their thermal evolution equilibriums range from 0.52 to 0.55 and 0.67 to 0.71, respectively [38]. For the oil samples analyzed, the values of $22S/(22S + 22R)$-$C_{31}H$ and $22S/(22S + 22R)$-$C_{32}H$ were 0.56–0.61 and 0.55–0.61, respectively (Table 1), which are both close to the equilibrium points. The $20S/(20S + 20R)$-$C_{29}S$ and $\beta\beta/(\alpha\alpha + \beta\beta)$-$C_{29}S$ values were 0.35–0.50 and 0.29–0.44, respectively, which are significantly lower than the equilibrium points (Table 1). These results indicate that the crude oils are in a low maturation stage.

### 5.2. Genetic Comparison

Based on cluster analysis of crude oil biomarkers and the comparison of source rocks, Sun [31] found that the $C_{35}H/C_{34}H$ and $G/C_{30}H$ values of oil from the Es4 source rocks were greater than 0.90 and 0.12, respectively, while the $C_{35}H/C_{34}H$ and $G/C_{30}H$ values of oil from the Es3 source rocks were less than 0.62 and 0.12, respectively. For these oil samples, the high-sulfur oils had higher $C_{35}H/C_{34}H$ and $G/C_{30}H$ values compared with the low-sulfur oils (Figure 6c,d). The $C_{35}H/C_{34}H$ and $G/C_{30}H$ values of the low-sulfur oils were generally in the range of 0.54–0.75 and 0.05–0.13, respectively. The values of the two high-sulfur oils were 0.82–1.11 and 0.12–0.23, respectively. These results suggest that the high-sulfur oils are mainly derived from Es4 source rocks, while the low-sulfur oils are mainly derived from Es3 source rocks.

In general, the precursors of $C_{27}$ regular steranes are derived primarily from plankton and certain algae, while the precursors of $C_{28}$ and $C_{29}$ regular steranes are derived primarily from phytoplankton and terrestrial plants, respectively [39,40]. Oil samples are generally clustered in a triangle diagram of $\alpha\alpha\alpha(20R)$-$C_{27}$, $C_{28}$, and $C_{29}$ regular steranes (Figure 6a), indicating that they have similar sources of parent material and are derived from both plankton and terrigenous plants.

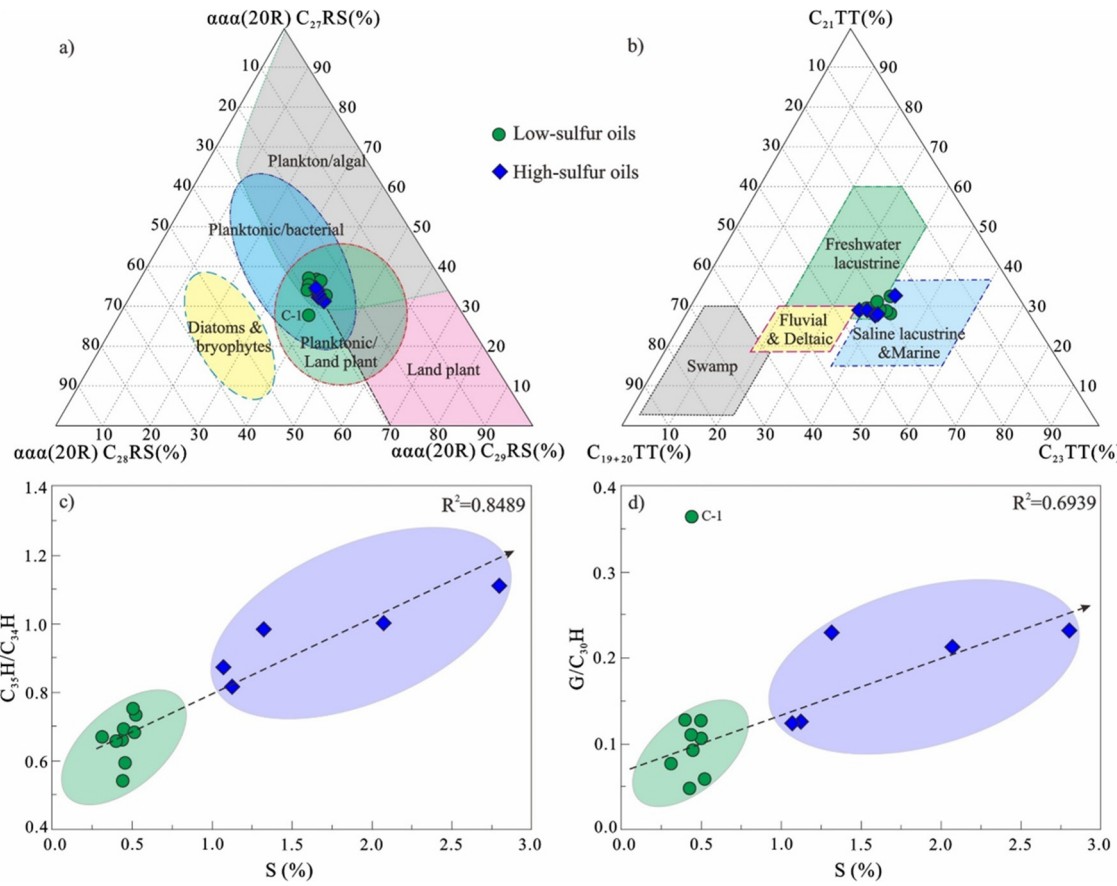

**Figure 6.** Diagrams showing the origin of the oil samples. (**a**) Triangle diagram of $\alpha\alpha\alpha$(20R) $C_{27}$, -$C_{28}$, -$C_{29}$ regular sterane (Huang and Meinschein, 1979 [39]); (**b**) Triangle diagram of $C_{19} + C_{20}$, $C_{21}$, and $C_{23}$TT (Xiao et al., 2019 [41]); (**c**) Cross plot of sulfur content versus $C_{35}$H/$C_{34}$H. (**d**) Cross plot of sulfur content versus G/$C_{30}$H.

The ratios of terpanes and hopanes with different carbon numbers can be used to distinguish sedimentary facies of the source rock. For example, based on the analysis of samples from around the world, lacustrine oil samples have higher $C_{26}$/$C_{25}$TT ratios (>1.1) and lower $C_{31}$R/$C_{30}$H ratios (<0.25) compared with most marine oil samples [35]. For the current oil samples, these two ratios are in the range 1.10–1.39 and 0.15–0.26 (Table 1), consistent with the characteristics of lacustrine oil. Thermal evolution and biodegradation have little effect on the distribution patterns of $C_{19}$–$C_{23}$ TT, and their relative contents can be used to distinguish crude oils of different origins [41]. In the triangle diagram of $C_{19} + C_{20}$, $C_{21}$, and $C_{23}$TT, the crude oil samples are closely distributed within the range of lacustrine oil (Figure 6b). However, the redox and salinity of the source rock deposition environment differ significantly between the low-sulfur and high-sulfur oils. Source rocks from anoxic reduction and high salinity environments are abundant in $C_{35}$H and gammacerane (G), which are characterized by higher $C_{35}$H/$C_{34}$H and G/$C_{30}$H content [42–44]. The $Es^3$ source rocks in the Miaoxi Sag have low G and $C_{35}$H contents; in contrast, the $Es^4$ source rocks deposited in a strong reduction and high salinity environment have high G and $C_{35}$H contents [30–32]. For these oil samples, high-sulfur oils have higher $C_{35}$H/$C_{34}$H and G/$C_{30}$H content compared with the low-sulfur oils (Figure 6c,d). This result suggests that high-sulfur oils have stronger associations with $Es^4$ source rocks. Sample C-1 had abnormally high G/$C_{30}$H content (Figure 6d) and deviated significantly from the other samples in the triangle diagram of $\alpha\alpha\alpha$(20R)-$C_{27}$, $C_{28}$, and $C_{29}$ steranes (Figure 6a). Given that intense biodegradation can significantly alter hopanes and steranes [19,20], the results likely infer that this oil has been severely biodegraded.

### 5.3. Degree of Biodegradation

Temperature is a crucial factor in biodegradation. It has been reported that the ideal temperature needed to sustain microbial life is below 80 °C, above which sterilization would occur in reservoirs [16,35,45]. The geothermal gradient in the eastern sea area of the Bohai Bay Basin is 26–42 °C/km, with a formation temperature of 52–84 °C at 2000 m [46,47]. Oil is produced at depths ranging from 1040 to 2600 m. The reservoir temperature is generally below 80 °C, and the formation water is mainly $CaCl_2$-type, with low to medium salinity [47,48], which provides favorable conditions for microorganism activity and reproduction [35]. The consumption of alkanes and the appearance of 25-norhopanes further support the biodegradation of these oils (Figures 2 and 3).

The PM scale is one of the most widely used methods for evaluating the level of biodegradation [15,19,20]. Biodegradation levels are categorized into scales ranging from PM0 (none) to PM10 (extreme) based on alterations of different compounds [35]. The biodegradation levels of the oil samples in this study varied from PM0 (sample F-1) to PM7 (sample C-1), as shown in Table 4, based on a comprehensive investigation of molecular geochemistry. Samples B-1 and B-2 were mixtures of slightly and severely biodegraded oils and were characterized by intact *n*-alkanes and the presence of 25-norhopane [25,35]. Given that later mixing with non-biodegraded or slightly biodegraded oils increases the concentration of $C_{30}H$ in the oil, the $C_{29}NH/C_{30}H$ ratio can be used to infer the level of biodegradation of the mixed oil. The biodegradation level of these two samples is estimated to be PM3.5 based on the $C_{29}NH/C_{30}H$ values (Figure 7).

**Table 4.** General characterization of saturated fractions of oil samples with different biodegradation levels (PM Scale).

| Types | PM Scale | General Characterization | Samples |
|---|---|---|---|
| No mixing | 0 | Intact *n*-alkanes (abundant light end *n*-alkanes) | F-1 |
| | 1 | Abundant *n*-alkanes, but light end *n*-alkanes consumed | H-1, G-1, G-2 |
| | 2 | *n*-Alkanes largely consumed; isoprenoids slightly consumed | D-2 |
| | 3 | *n*-Alkanes severely consumed; isoprenoids largely consumed | E-1 |
| | 4 | *n*-Alkanes removed; isoprenoids severely consumed, but no 25-norhopanes | Not found in this study |
| | 5 | Isoprenoids removed; 25-norhopanes present, but 25-norhopanes no more than hopanes in *m/z* 177 | D-1, E-2 |
| | 6 | Abundant 25-norhopanes; 25-norhopanes exceed hopanes in *m/z* 177 | A-1, A-2, A-3 |
| | 7 | Hopanes almost completely converted to 25-norhopanes; regular steranes significantly altered | C-1 |
| Mixing | ~3.5 | Abundant *n*-alkanes and isoprenoids; 25-norhopanes present | B-1, B-2 |

Note: The PM Scale of samples B-1 and B-2 was inferred based on their $C_{29}$ 25-norhopane/$C_{30}$ hopanes values. The PM scale is based on previous studies ([19,20,35]).

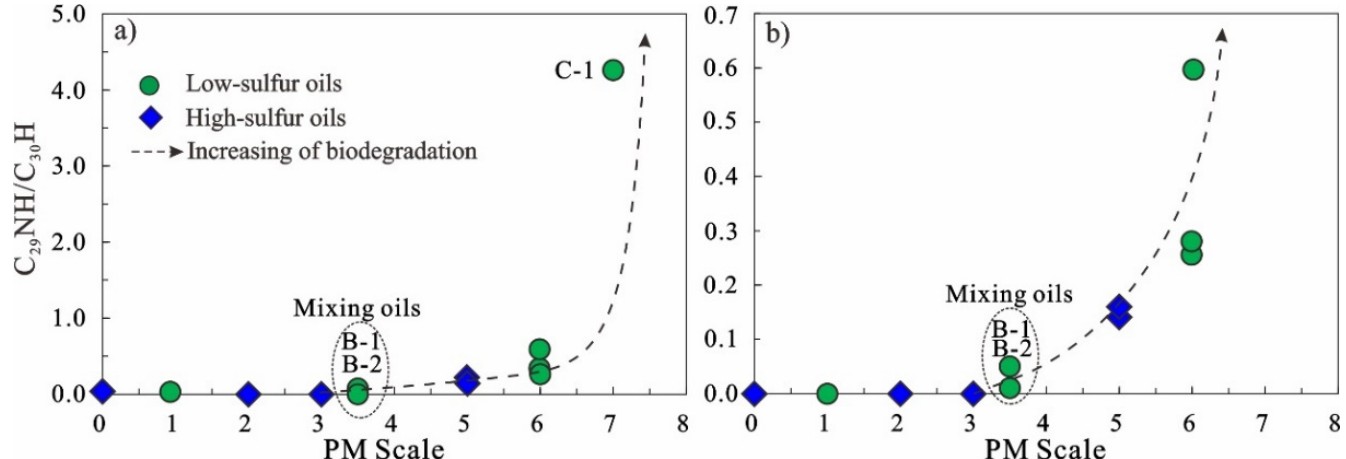

**Figure 7.** Cross plots of the degree of biodegradation (PM Scale) versus the $C_{29}NH/C_{30}H$ biomarker ratio. (**b**) Partial enlargement of (**a**).

### 5.4. Effect of Biodegradation on Inorganic Elements and Element Ratios

Although the concentrations of the major and trace elements in different oil samples varied by orders of magnitude, there were no significant differences in the concentrations of elements in the low-sulfur oils and high-sulfur oils. The major and trace element distribution curves for the high-sulfur oils were comparable with those of the low-sulfur oils (Figures 4b and 5b). Given that high-sulfur oil reflects a more reductive and salty sedimentary environment, this result suggests that the concentrations of most major and trace elements in these oil samples are less affected by the sedimentary environment of the source rocks.

It should be noted that the concentrations of major elements in oil samples are closely associated with the biodegradation level and the major elements show different trends as the biodegradation level increases (Figure 8): Mg, Ca, Mn, and Fe concentrations increase significantly (defined as Group A); the concentrations of Na, K, Ti, and Al only show a remarkable increase during the intense biodegradation stage (PM ≥ 4) (defined as Group B); and the concentration of non-metal P is stable (defined as Group C). Similar to the major elements, trace elements can also be categorized into three groups based on the trends of their biodegradation processes: Be, Sc, Rb, Sr, Zr, Pb, Th, and U are classified into Group A; Cr, Zn, Cs, Nb, Ba, Hf, and Tl are classified into Group B; and Li, V, Co, Ni, Cu, Ga, Sn, and Ta are classified in Group C. The concentrations of P, Co, and Ni remain stable during the biodegradation process, while the concentrations of Zn and Ba increase significantly only at the PM ≥ 4 stage. As shown in Figure 9, the nine trace elements display three distinct patterns. The results suggest that major and trace elements are differentially enriched during the biodegradation process.

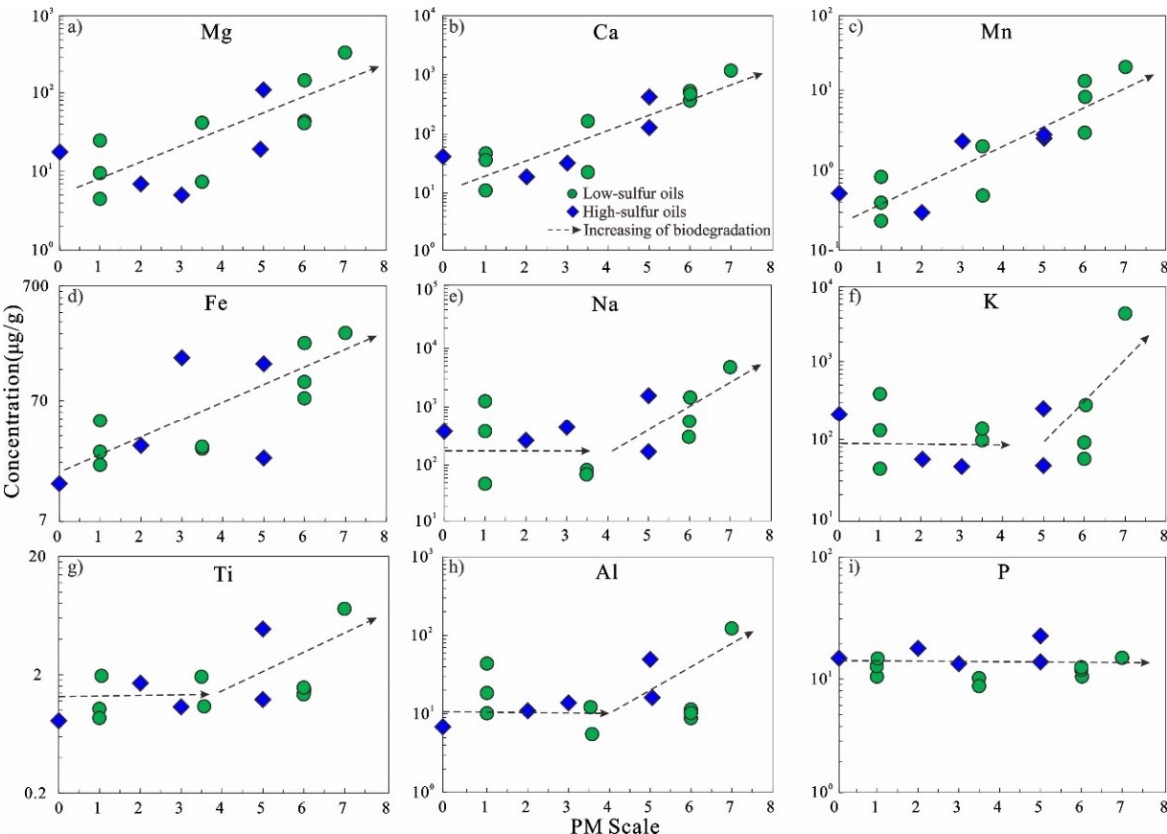

**Figure 8.** Cross plots showing differential enrichment of major elements based on increasing biodegradation levels (PM scale). (**a**) Cross plot of PM Scale versus Mg; (**b**) Cross plot of PM Scale versus Ca; (**c**) Cross plot of PM Scale versus Mn; (**d**) Cross plot of PM Scale versus Fe; (**e**) Cross plot of PM Scale versus Na; (**f**) Cross plot of PM Scale versus K; (**g**) Cross plot of PM Scale versus Ti; (**h**) Cross plot of PM Scale versus Al; (**i**) Cross plot of PM Scale versus P.

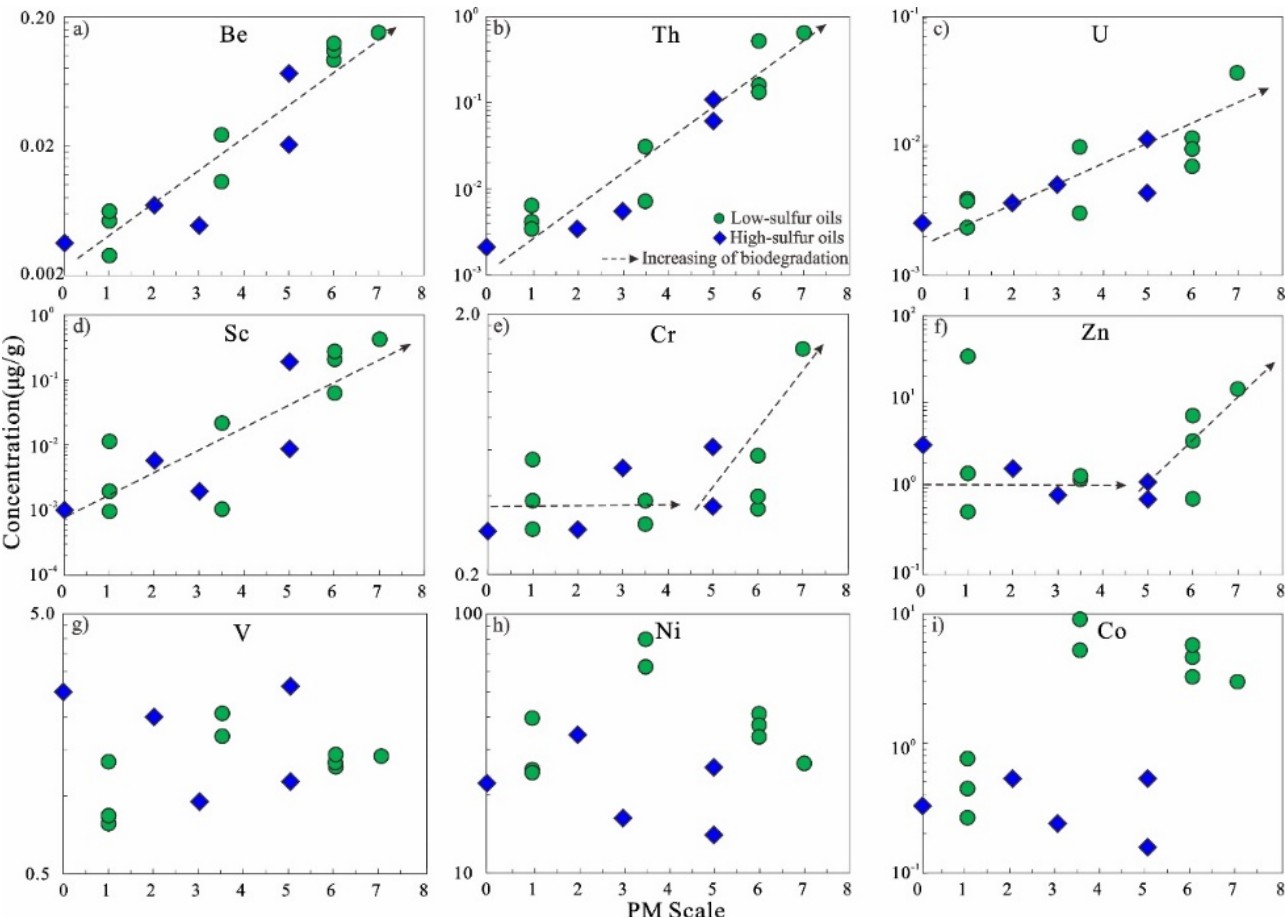

**Figure 9.** Cross plots showing differential enrichment of trace elements based on increasing biodegradation levels (PM scale). (**a**) Cross plot of PM Scale versus Be; (**b**) Cross plot of PM Scale versus Th; (**c**) Cross plot of PM Scale versus U; (**d**) Cross plot of PM Scale versus Se; (**e**) Cross plot of PM Scale versus Cr; (**f**) Cross plot of PM Scale versus Zn; (**g**) Cross plot of PM Scale versus V; (**h**) Cross plot of PM Scale versus Ni; (**i**) Cross plot of PM Scale versus Co.

Cluster analysis of element concentration is commonly used in the classification of geological samples [3,10,12]. There are similarities between samples within the same class, and discrepancies between samples in different classes. In this study, a systematic clustering approach was implemented in IBM SPSS Statistical software (version 22.0). The data were normalized using the standard deviation method and Euclidean distance was chosen as the standard for similarity measurements. The results show that oil samples can be classified into type A and type B, and type A oils can be divided further into type A1 and type A2 based on elements in Group A. As shown in Figure 10, the type A1 oils are characterized by a PM scale of 0–5 and $C_{29}NH/C_{30}H$ values of 0–0.14; type A2 oils are in a higher PM scale of 5–6 and the $C_{29}NH/C_{30}H$ values are in the range 0.16–0.59. Type B oils are the most highly biodegraded, with a PM scale of 7 and a $C_{29}NH/C_{30}H$ value of 4.26.

The V/Ni ratio can reflect reduction in water during the deposition of source rocks [49,50]; V/Ni = 1 was used as a criterion for separating marine from continental crude oils [3,51]. The ratios of other elements, including V/Co, Ni/Co, Cr/V, Sc/V, and Th/U, are also frequently used in the reconstruction of sedimentary environments and are considered potential oil–oil (source) correlation parameters [3,11,12,52]. As shown in Figure 11, biodegradation has a different effect on these ratios. As biodegradation increases, the V/Ni, V/Co, and Ni/Co ratios initially decrease and then increase, while the Cr/V, Sc/V, and Th/U ratios initially remain unchanged before increasing rapidly. In particular, the Sc/V and Th/U ratios show a significant increase when PM is ≥4.

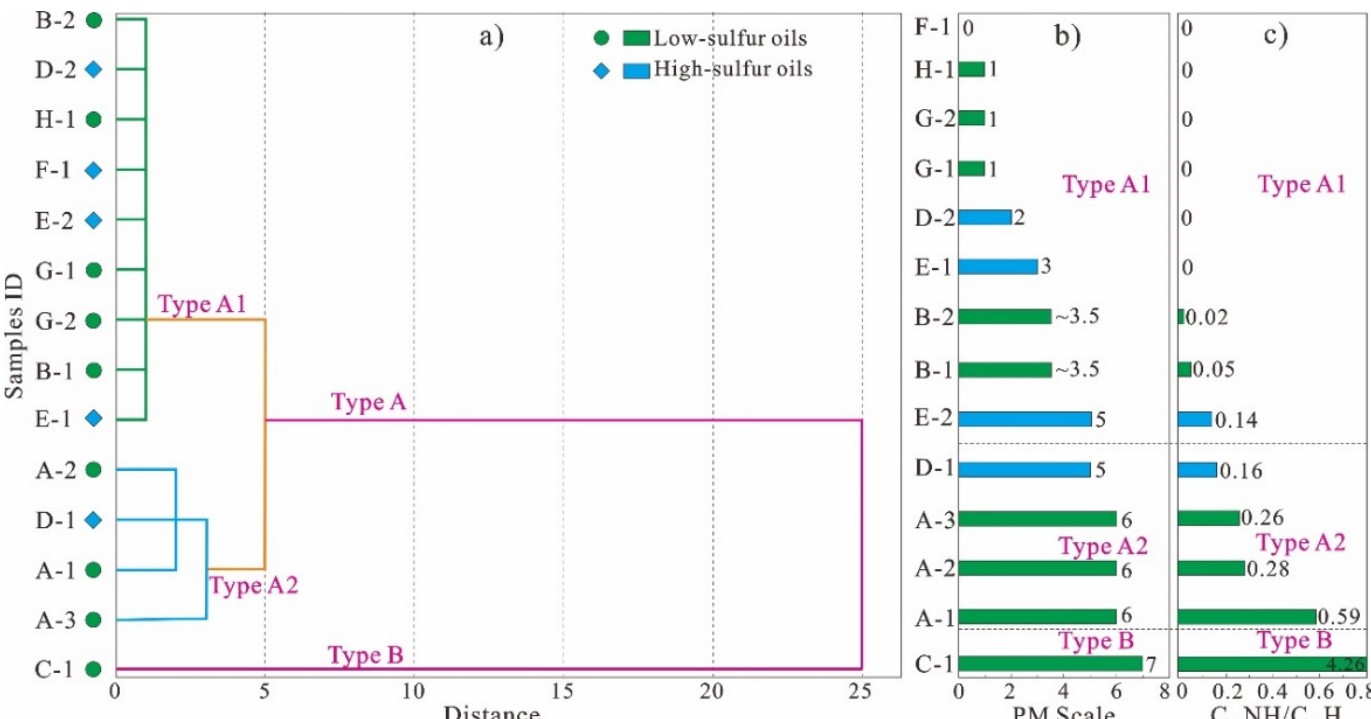

**Figure 10.** Dendrogram showing cluster analysis of the concentrations of Mg, Ca, Mn, Fe, Be, Sc, Rb, Sr, Zr, Pb, Th, and U (**a**), and histograms of PM Scale (**b**) and $C_{29}NH/C_{30}H$ (**c**).

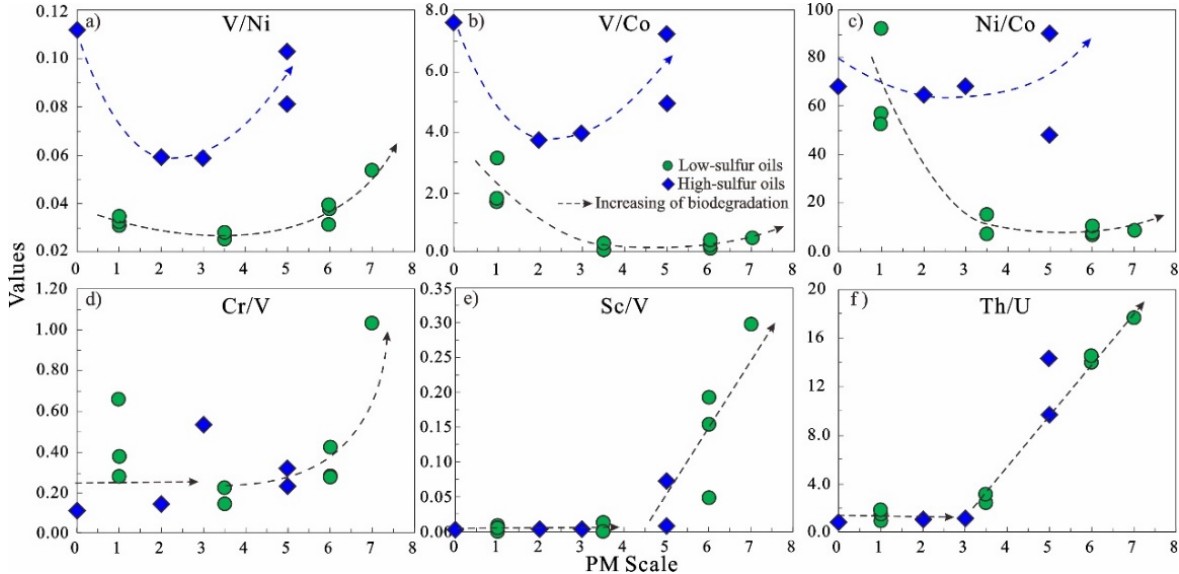

**Figure 11.** Cross plots showing changes in the ratios of some trace elements based on increasing biodegradation levels (PM scale). (**a**) Cross plot of PM Scale versus V/Ni; (**b**) Cross plot of PM Scale versus V/Co; (**c**) Cross plot of PM Scale versus Ni/Co; (**d**) Cross plot of PM Scale versus Cr/V; (**e**) Cross plot of PM Scale versus Sc/V; (**f**) Cross plot of PM Scale versus Th/U.

Based on the different behaviors of these elements during biodegradation, several ratios, including Mg/P, Ca/P, Mn/P, and Fe/P, can be used to infer the biodegradation level; higher values indicate greater biodegradation (Figure 12). This is also reflected in the relationship between these ratios and the density of crude oil; increased biodegradation results in an increase in the density of crude oil. The densities of the oil samples were positively correlated with these ratios (Figure 13), inferring that these ratios may, to some extent, indicate biodegradation. This result is also reflected in oil samples from other areas.

For example, oil from the Chepaizi Bulge in the western margin of the Junggar Basin has undergone biodegradation to varying degrees, and previous studies have shown that the density of this oil increases with increasing levels of biodegradation [53,54]. The densities of the oils in this study were positively correlated with the ratios of Mg/P, Ca/P, Mn/P, and Fe/P, as shown in Figure 14.

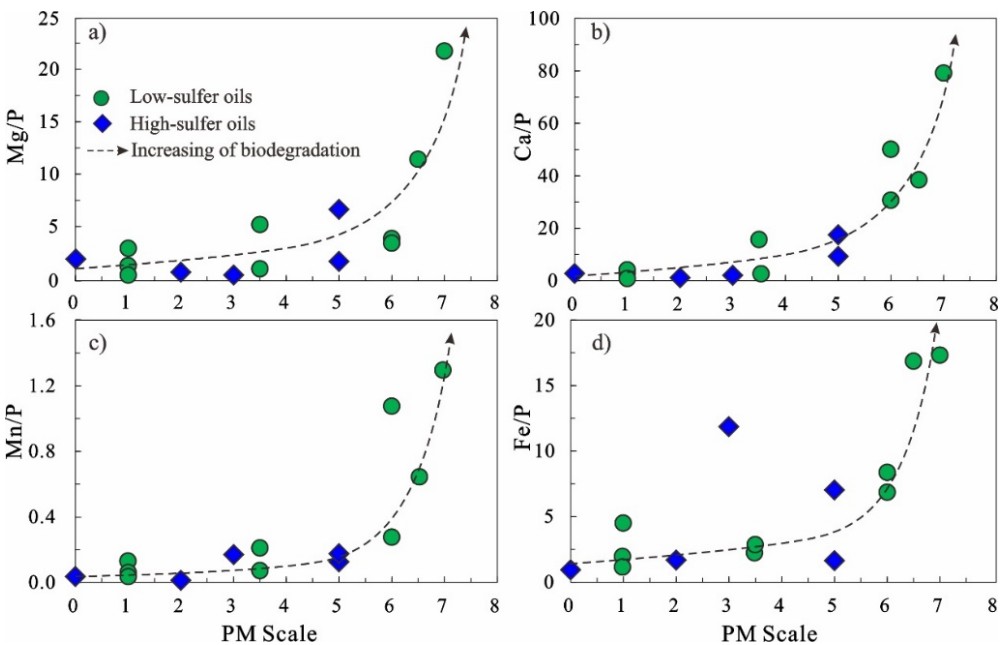

**Figure 12.** Cross plots of the degree of biodegradation (PM Scale) versus Mg/P (**a**), Ca/P (**b**), Mn/P (**c**) and Fe/P (**d**) ratios.

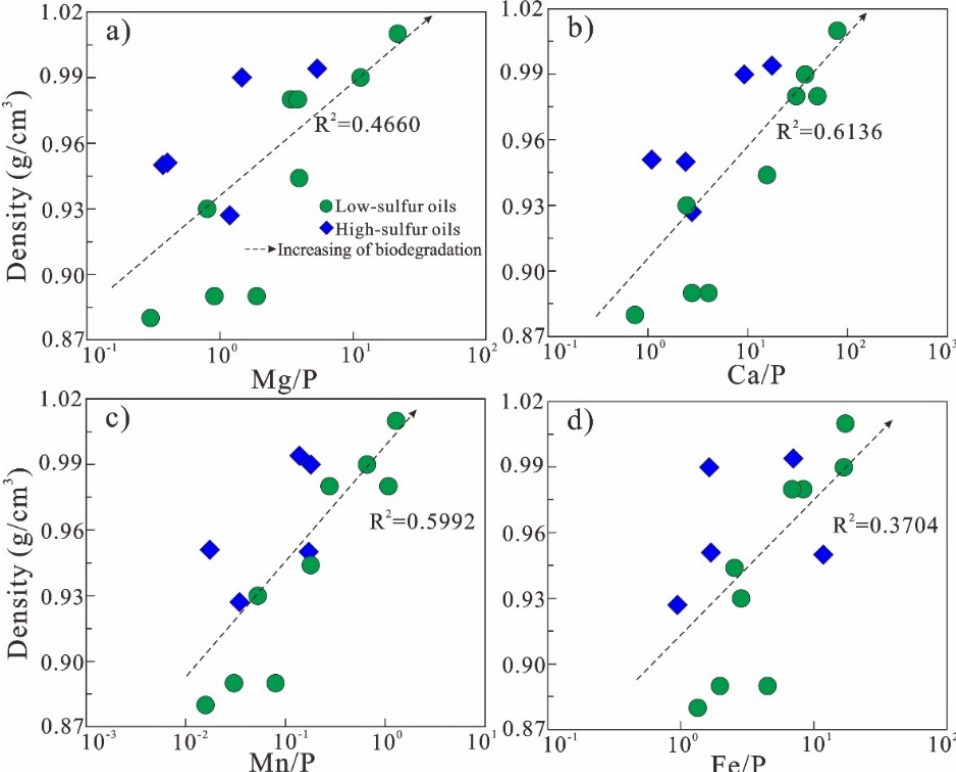

**Figure 13.** Cross plots of Mg/P (**a**), Ca/P (**b**), Mn/P (**c**) and Fe/P (**d**) ratios versus the densities of oil samples in this study.

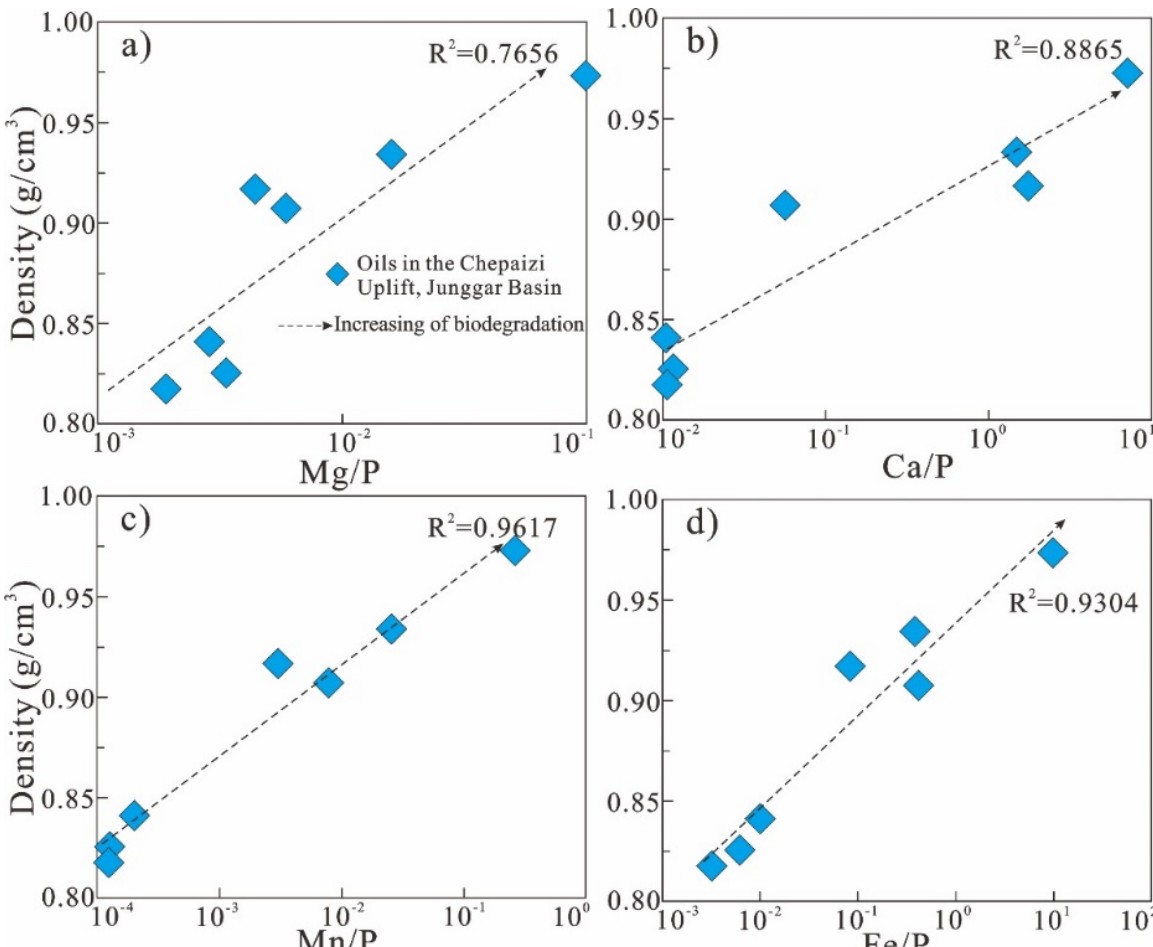

**Figure 14.** Cross plots of Mg/P (**a**), Ca/P (**b**), Mn/P (**c**) and Fe/P (**d**) ratios versus densities of oil samples from the Chepai Bulge in the western margin of the Junggar Basin.

There were considerable variations in V/Ni, V/Co, and Ni/Co values between low-sulfur oils and high-sulfur oils, but there were no distinctions between Cr/V, Sc/V, and Th/U values in the two groups (Figure 11). This result suggests that the ratios, including V/Ni, V/Co, and Ni/Co, may be used to distinguish high-sulfur oil from low-sulfur oil, given that the concentrations of V, Ni, and Co are not correlated with biodegradation levels.

### 5.5. Differential Enrichment of Inorganic Elements during Biodegradation

The metal elements in crude oil exist mainly in the form of inorganic salts, organic acid salts, porphyrins, and non-porphyrins [1,4]. It is generally believed that as the hydrocarbon components are consumed with increasing biodegradation, these compounds become relatively enriched, resulting in an increase in the concentration of metal elements [19,20].

This phenomenon may be associated with the effect of biodegradation on the oil–water–rock interaction in reservoirs. Complex oil–water–rock interactions occur during crude oil migration and accumulation [55–59]. Inorganic elements in petroleum can be derived from water and reservoir rocks in addition to source rocks [2,12]. Inorganic elements in crude oil, including Ca, Na, K, and Mg, are likely to be affected by formation water, since $CaCl_2$-type formation water in the study area contains various ions, including $Ca^{2+}$, $Na^+$, $K^+$, $Mg^{2+}$, $Cl^-$, $HCO^{3-}$, $SO_4^{2-}$, and $CO_3^{2-}$ [48]. However, the patterns of Ca and Mg distribution in crude oil samples are similar to those of Mn and Fe, while the patterns of Na and K distribution are similar to that of Al (Figure 8), suggesting that the influence of formation water on these elements in oil samples is limited.

The minerals in rocks are a major source of inorganic elements. For example, the major elements, including Al, Na, K, and Ti, are closely linked to various silicate minerals such as feldspar and clay minerals [60,61]; Ca and Mg can be present in silicate minerals, but are primarily concentrated in carbonate rock minerals such as calcite and dolomite [12]. Fe and Mn originate from a variety of sources, including hydrothermal deposits [62,63] and volcanic, silicate, and carbonate minerals [2,64]. Here, the clastic reservoirs in the study area are rich in feldspar clasts and volcanic debris, and the cements are mainly carbonate minerals, followed by clay minerals [65,66]. Elements in the minerals are likely extracted into the crude oil once the crude oil interacts with minerals in these reservoirs.

The biodegradation process can affect the interaction between crude oil and reservoir minerals because it generates an acidic substance. Crude oil becomes more acidic as it undergoes biodegradation [67,68]. For example, the total acid number (TAN) in lacustrine oils from the Muglad and Melut basins in Sudan and the Bohai Bay basin in China increases with increasing levels of biodegradation, especially when PM $\geq$ 4 (Figure 15). Numerous studies have revealed that organic acids formed during biodegradation can affect the TAN [69]. Biodegradation converts long-chain acids into short-chain acids [12]. Thus, such organic acids might be similar to the organic acids released during thermal evolution of source rocks and can dissolve reservoir minerals and extract inorganic elements, resulting in differential enrichment of elements in crude oil. On the other hand, the organic acid salts generated during biodegradation can provide a large number of carriers for metal elements.

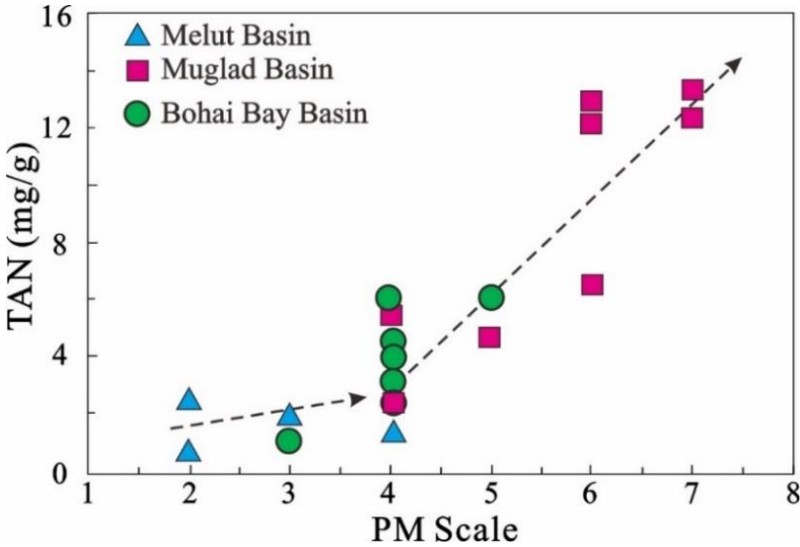

**Figure 15.** Cross plot of the degree of biodegradation (PM Scale) versus total acid number (TAN). Data from Dou [70].

The model for differential enrichment of major elements during biodegradation is summarized as follows (Figure 16): (1) Changes in Ca, Mg, Fe, and Mn follow the same trend during the biodegradation process. Carbonate cements in reservoirs produce Ca, Mg, Fe, and Mn. Because carbonate is sensitive to organic acids, the small quantities of organic acids generated at the beginning of biodegradation react preferentially with the carbonate minerals, and Ca, Mg, Fe, and Mn are extracted from these minerals into the oil. (2) Na, K, Al, and Ti generally show consistent patterns during the biodegradation process. These elements are present primarily in silicate minerals like feldspar, which are less sensitive to organic acid compared with carbonate minerals. However, when large amounts of organic acids are produced during the intense biodegradation stage (PM $\geq$ 4), silicate minerals are also affected, and Na, K, Al, and Ti can be extracted from these minerals and into the oil. (3) P commonly exists in petroleum in a non-porphyrin organic form [12] and promotes biodegradation [71,72]. P levels remain constant in oil samples, likely due to the high preservation of P during the biodegradation process. P can be preserved in the sediments

through organic binding and will diffuse from the sediment into the water body under anoxic reduction conditions [73,74]. The Fe redox cycle in sediments can precipitate P into authigenic phosphate minerals [74]. In short, the P reaction is weaker than that of other metal elements during the complex oil–water–rock reactions and its absolute concentration in oil can thus remain relatively stable.

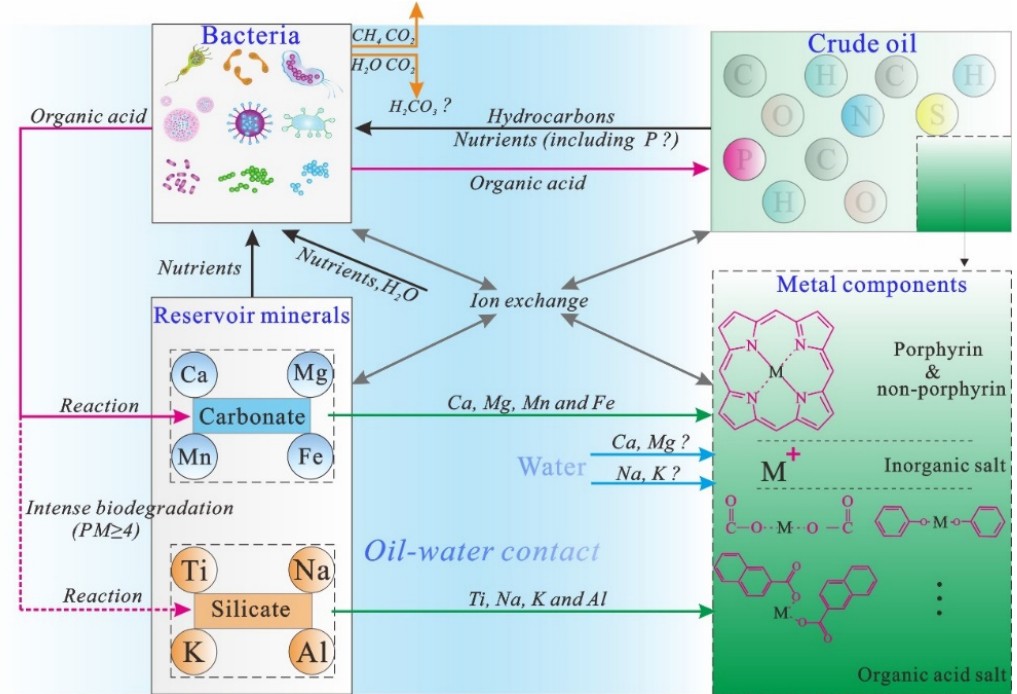

**Figure 16.** A model showing the possible enrichment mechanism of major elements in oil during the biodegradation process.

## 6. Conclusions

The molecular markers and the major and trace elements in 14 crude oil samples from the centralto southern Miaoxi Sag in the Bohai Bay Basin were analyzed, and the effect of the progress of biodegradation on the major and trace elements investigated. The following conclusions were reached:

(1)   The lower $20S/(20S + 20R)$-$C_{29}S$ and $\beta\beta/(\alpha\alpha + \beta\beta)$-$C_{29}S$ values indicate that the crude oil samples have low maturity. Comparisons of several genetic parameters suggest that these oils have similar sources of parent material and are contributed by both plankton and terrigenous plants. Their source rocks develop in a lacustrine environment, but high-sulfur oils are derived from source rocks that are more reductive and saltier than low-sulfur oils. The application of biomarker compounds and trace elements confirmed that high-sulfur and low-sulfur oils have different sources of parent material. A set of hydrocarbon source rocks with strong reduction and brackish water deposits exist in the Miaoxi Sag; this set of hydrocarbon source rocks may offer a favorable direction for future oil and gas explorations in the Miaoxi Sag of the Bohai Bay Basin.

(2)   The concentrations of Mg, Ca, Mn, Fe, Be, Sc, Rb, Sr, Zr, Pb, Th, and U increase with increasing biodegradation levels. The concentrations of Na, K, Ti, Al, Cr, Zn, Cs, Nb, Ba, Hf, and Tl show remarkable results only during the intense biodegradation stage (PM $\geq$ 4), while the concentrations of P, Li, V, Co, Ni, Cu, Ga, Sn, and Ta are not affected by biodegradation. Biodegradation affects the ratios of V/Ni, V/Co, Ni/Co, Cr/V, Sc/V, Th/U, with the Sc/V and Th/U ratios increasing significantly when PM is $\geq$4. Some ratios, including those of Mg/P, Ca/P, Mn/P, and Fe/P, are proposed as favorable indicators of biodegradation. Differential enrichment of inorganic elements

may be associated with the influence of organic acids on the oil–water–rock reservoir interactions during the biodegradation process.

**Author Contributions:** Conceptualization, H.Y. and D.W.; methodology, F.W.; validation, Y.G. and G.T.; formal analysis, P.S.; investigation, P.S.; writing—original draft preparation, H.Y. and P.S.; writing—review and editing, Y.T.; project administration, Y.T. All authors have read and agreed to the published version of the manuscript.

**Funding:** This study was financially supported by the National Natural Science Foundation of China (No. 42202163), and the China Postdoctoral Science Foundation (No. 2022M710488).

**Data Availability Statement:** Data is contained within the article. The data presented in this study are available in Tables 1 and 2.

**Conflicts of Interest:** The authors declare no conflict of interest.

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
