# Peer review of "Differential Enrichment of Trace and Major Elements in Biodegraded Oil: A Case Study from Bohai Bay Basin, China"

_processes, doi:10.3390/pr11041176_

Round 1

Reviewer 1 Report

This manuscript investigates the differential enrichment of elements during the process of oil biodegradation based on the analysis of 14 biomarker compounds of crude oil and the main amount of trace elements. It is generally well written and organized but more clarification and language improvement is still required. I would like recommend minor revision before publication.

1. Some of the language in the manuscript are problematic, and it is recommended that language should be checked carefully throughout the text and every symbol should be standardized. Such as Line 326, Line246-248

2. Relevant explanations and additions to the drawing are needed, such as Fig1.A lack of the plotting scale.

3. Cao et.al (2007) proposed that stratigraphic effects may have some influence on elements, and the effect of transport on elements is not considered in the manuscript, please add relevant evidence.

Cao, J., Hu, W., Yao, S., Zhang, Y., Wang, X., Zhang, Y., Tang, Y., Shi, X., 2007. Mn content of reservoir calcite cement: A novel inorganic geotracer of secondary petroleum migration in the tectonically complex Junggar Basin (NW China). Science in China (Series D: Earth Sciences) 12, 1796–1809.

4. The sources of high and low sulfur oils are not described in the manuscript, and the biodegradation grades is not clear enough; additional evidence is recommended. Please Read Chen et.al 2016. (doi.org/10.1016/j.precamres.2018.08.005).

Author Response

Reviewer1

Comment

Some of the language in the manuscript are problematic, and it is recommended that language should be checked carefully throughout the text and every symbol should be standardized. Such as Line 326, Line246-248

Response

We appreciate the reviewer’s comment and we have rewrite this sentence. In addition, we have touched up the language of the manuscript to ensure that it is acceptable.

Comment

Relevant explanations and additions to the drawing are needed, such as Fig1.A lack of the plotting scale.

Response

We checked all the drawings in the manuscript. We have modified Fig.1 and added the plotting scale (see line 945).

 Comment

Cao et.al (2007) proposed that stratigraphic effects may have some influence on elements, and the effect of transport on elements is not considered in the manuscript, please add relevant evidence.

Response

First of all, we agree with the reviewer’s point that the effects of transport should be considered.

However, we can see from Figure 1 that the distribution of sample points is relatively concentrated and the distances between them are short; meanwhile, the crude oils in the study area are all nearby hydrocarbon source rocks and do not experience long-distance transport(Fig.1). Thus, the effect from transport is not the key factor controlling the concentrations of trace elements in this case.

Comment

The sources of high and low sulfur oils are not described in the manuscript, and the biodegradation grades is not clear enough; additional evidence is recommended. Please Read Chen et.al 2016. (doi.org/10.1016/j.precamres.2018.08.005).

Response

We appreciate the reviewer’s kind reminds and critical points. Based on the reviewers' comments, we have added the following statement to the second paragraph of Section 5.2, lines196-205:

Based on cluster analysis of crude oil biomarkers and the comparison of source rocks, Sun et al., (2021) found the C35H/C34H and G/C30H values of the oil from the Es4 source rocks are greater than 0.90 and 0.12, respectively; the C35H/C34H and G/C30H values of the oil from the Es3 source rocks are less than 0.62 and 0.12, respectively. For these oil samples, compared with the low-sulfur oils, the high-sulfur oils show higher C35H/C34H and G/C30H values (Fig. 6c and d). The C35H/C34H and G/C30H values of the low-sulfur oils are generally in the range of 0.54–0.75 and 0.05–0.13, respectively; those two values for high-sulfur oils are in the range of 0.82–1.11 and 0.12–0.23, respectively. This result suggests that the high-sulfur oils are mainly derived from Es4 source rocks, while the low-sulfur oils are mainly derived from Es3 source rocks.

Reviewer 2 Report

This study discussed the influence of biodegradation on the distribution of trace and major elements in crude oil.  The authors also mentioned that inorganic elements in petroleum may be derived from water and reservoir rocks besides the source rocks. Meanwhile, for biodegraded oil, it seems the consuming of hydrocarbons will lead to the increase of metal element concentration. Therefore, the absolute concentration of a certain elements in crude oil should be controlled by at least two factors, outside input (water and surrounding rock) and the changes inside. Actually, most of the biodegradation activities occur at the oil-water interface. It seems the living microorganism also can bring some elements into the crude oil from the reservoir water during microbial activities. The difference of absolute concentration for essential elements between microbes and crude oil might lead to quite different influence of biodegradation on the distribution of trace and major elements in crude oil. 

Previous studies have pointed out some elements related to the life activities, such as N, P, K, Cu, Ni, Zn, Ba. Their evolution trends should be reported and discussed in the revised edition.

In Part 5.4, the authors mentioned that “Given that high-sulfur oil reflects a more reductive and salty sedimentary environment, this result suggests that the concentrations of most major and trace elements in these oil samples are less affected by the sedimentary environment of the source rocks.” However based on the data shown in Fig. 4b and 5b, the difference in the absolute element concentrations between high-sulfur oil and low –sulfur oil are huge (an order of magnitude difference). It is little visual difference in the Fig.4b and 5b, because the Y-axis is using logarithmic coordinate.

Author Response

Reviewer2

 Comment:

Previous studies have pointed out some elements related to the life activities, such as N, P, K, Cu, Ni, Zn, Ba. Their evolution trends should be reported and discussed in the revised edition.

Response:

We appreciate the reviewer’s comment. We have discussed the evolution of the relevant elements in Section 5.4, lines 278-287 as follows: “the concentrations of Mg, Ca, Mn, and Fe increase dramatically (defined as Group A); the concentrations of Na, K, Ti, and Al only display a remarkable increase in the intense bio-degradation stage (PM≥ 4) (defined as Group B); specifically, the concentration of nonmetal P is stable (defined as Group C). Similar to major elements, trace elements can also be categorized into three groups based on their trends in the biodegradation process: Be, Sc, Rb, Sr, Zr, Pb, Th, and U are classified in the Group A; Cr, Zn, Cs, Nb, Ba, Hf, and Tl are classified in the Group B; Li, V, Co, Ni, Cu, Ga, Sn, and Ta are in the Group C.” In other words, the concentrations of P, Co and Ni remain stable during the biodegradation process, while the concentrations of Zn and Ba increase dramatically only at the stage of PM≥ 4. In the Section 5.5, lines 404-411, we used major elements as a model to showing the differential enrichment of these elements. For example, we discussed that “P in oil samples remains constant, and this may be due to the well preservation of P in the biodegradation process. P can be preserved in the sediments in the form of organic binding, and will diffuse from the sediment to the water body under the condition of anoxic reduction (Benitez-Nelson, 2000; Algeo and Ingall, 2007). The redox cycle of Fe in sediments can precipitate P into authigenic phosphate minerals (Algeo and Ingall, 2007). In short, during the complex oil-water-rock reactions the reaction of P is weaker than that of other metal elements, thus its absolute concentration in oil can remain relatively stable.”

Comment

In Part 5.4, the authors mentioned that “Given that high-sulfur oil reflects a more reductive and salty sedimentary environment, this result suggests that the concentrations of most major and trace elements in these oil samples are less affected by the sedimentary environment of the source rocks.” However based on the data shown in Fig. 4b and 5b, the difference in the absolute element concentrations between high-sulfur oil and low –sulfur oil are huge (an order of magnitude difference). It is little visual difference in the Fig.4b and 5b, because the Y-axis is using logarithmic coordinate.

 Response

we agree with the reviewer’s point. However, we just want to show that the distribution of most of the elemental concentrations of high sulfur oil is within the distribution of low sulfur oil, which is more obvious in Figure 8 and Figure 9. In other words, it is not possible to distinguish well between high and low sulfur oils using these elemental concentrations. The crude oil in this study suffered from biodegradation, and this phenomenon may be caused by biodegradation. However, for the normal undegraded crude oil, the difference in deposition environment might be reflected in the difference in elemental concentrations. Considering that this study is mainly focused on the effect of biodegradation on elements and to avoid misunderstanding to the readers, we have deleted this paragraph statement.

Reviewer 3 Report

In this manuscript, the authors reported that the case study with 14 crude oils from the Miaoxi Sag of the Bohai Bay Basin, eastern 12 China were analyzed with molecular markers, trace and major elements to investigate the effect of 13 biodegradation on the inorganic element. The authors claimed that the differential enrichment of these elements is related to the effect of organic acids generated by biodegradation on the reservoir oil-water-rock interactions. In overall, this manuscript is interesting but in order to consider publication, this work should be revised. The following comments should be addressed for the improvement of their manuscript.

Comment 1: The overall study aims for this the differential enrichment of trace and major elements in biodegraded oil using organic molecular markers need to be further clarified in detail as compared to other conventional system. The advantages and benefits of organic molecular markers in the present study need to be clarified too.

Comment 2: The authors should provide a fair and complete literature review about the previous works in the differential enrichment of elements in the biodegradation process including the maturity, origin, and biodegradation level of the oil samples should be summarized in a table as benchmarking purpose and discussed in detail with your research findings.

Comment 3: The future direction and perspectives for the crude oils from Bohai Bay Basin, China after the case study and evaluation can be further discussed in detail in the conclusion section.

Comment 4: The carefully English correction is necessary for the whole manuscript. Please check and revise accordingly.

Author Response

Reviewer3

Comment

The overall study aims for this the differential enrichment of trace and major elements in biodegraded oil using organic molecular markers need to be further clarified in detail as compared to other conventional system. The advantages and benefits of organic molecular markers in the present study need to be clarified too.

 Response

We appreciate the reviewer’s kind reminds and critical points. Based on the reviewers' comments, we have added the following statement to the second paragraph of Section 5.2, lines 196-205:

Based on cluster analysis of crude oil biomarkers and the comparison of source rocks, Sun et al., (2021) found the C35H/C34H and G/C30H values of the oil from the Es4 source rocks are greater than 0.90 and 0.12, respectively; the C35H/C34H and G/C30H values of the oil from the Es3 source rocks are less than 0.62 and 0.12, respectively. For these oil samples, compared with the low-sulfur oils, the high-sulfur oils show higher C35H/C34H and G/C30H values (Fig. 6c and d). The C35H/C34H and G/C30H values of the low-sulfur oils are generally in the range of 0.54–0.75 and 0.05–0.13, respectively; those two values for high-sulfur oils are in the range of 0.82–1.11 and 0.12–0.23, respectively. This result suggests that the high-sulfur oils are mainly derived from Es4 source rocks, while the low-sulfur oils are mainly derived from Es3 source rocks.

Comment

The authors should provide a fair and complete literature review about the previous works in the differential enrichment of elements in the biodegradation process including the maturity, origin, and biodegradation level of the oil samples should be summarized in a table as benchmarking purpose and discussed in detail with your research findings.

Response

We appreciate the reviewer’s kind reminds and critical points. Inorganic elements are thought to be mainly endowed in heavy fractions such as pring, and therefore the depletion of hydrocarbon fractions by biodegradation was considered by previous authors to result in a relative enrichment of inorganic elements. Perhaps due to this "consensus", previous studies have focused on the effects of maturation and deposition environment on inorganic elements, but not on the differential enrichment elements in the biodegradation. We have only found Akinlua (2007) reported the concentrations of the metals, such as Co, Cu, Fe, Ni and V, were more in bio-degraded oils than non-degraded oils except Cr. The study did not provide the level of biodegradation of the samples. Therefore, the present study is somewhat original and first of its kind. This means that it is very difficult and impractical to provide a complete and comprehensive literature review as you suggested.

Akinlua, A., Ajayi, T., Adeleke, B., 2007. Organic and inorganic geochemistry of northwestern Niger Delta oils. Geochemical Journal 41, 271–281.

Comment

The future direction and perspectives for the crude oils from Bohai Bay Basin, China after the case study and evaluation can be further discussed in detail in the conclusion section.
Response

We appreciate the reviewer’s kind reminds and critical points. We have added this section to the discussion(see line 429-434).

The application of biomarker compounds and trace elements confirmed that high and low sulfur oils have different parent material sources. A set of hydrocarbon source rocks with strong reduction and brackish water deposition exists in the Miaoxi Depression, and this set of hydrocarbon source rocks can be a favorable direction for the next oil and gas exploration in the Miaoxi Depression of the b Bohai Bay Basin.

Comment

The carefully English correction is necessary for the whole manuscript. Please check and revise accordingly

Response

We appreciate the reviewer’s kind reminds and critical points. We have touched up the language of the manuscript to ensure that it is acceptable.

Round 2

Reviewer 3 Report

In overall, this manuscript was technically well revised. This revised manuscript meets the criteria of processes. Therefore, in my opinion, the revised manuscript can be accepted for publication.